nanotechnology/chemical biology/biochemistry

gold nanorods, microbubbles, photothermal therapy, chemotherapy

**Author for correspondence:**
Jinliang Liu
e-mail: liujl@shu.edu.cn

This article has been edited by the Royal Society of Chemistry, including the commissioning, peer review process and editorial aspects up to the point of acceptance.

# Chemodrug-gated mesoporous nanoplatform for new near-infrared light controlled drug release and synergistic chemophotothermal therapy of tumours

Cuiping Fu, Jialin Lu, Yihan Wu, Yong Li and Jinliang Liu

School of Environmental and Chemical Engineering, Shanghai University, Shanghai 200444, People's Republic of China

(iD) JL, 0000-0003-1688-5969

Controlled drug release and synergistic therapies have an important impact on improving therapeutic efficacy in cancer theranostics. Herein, a new near-infrared (NIR) light-controlled multi-functional nanoplatform (GNR@mSiO$_2$-DOX/PFP@PDA) was developed for synergistic chemo-photothermal therapy (PTT) of tumours. In this nano-system, doxorubicin hydrochloride (DOX) and perfluoro-$n$-pentane (PFP) were loaded into the channels of mesoporous SiO$_2$ simultaneously as a first step. A polydopamine (PDA) layer as the gatekeeper was coated on their surface to reduce premature release of drugs at physiological temperature. Upon 808 nm NIR irradiation, the gold nanorods (GNR) in the core of the nanoplatform show high photothermal conversion efficiency, which not only can provide the heat for PTT, but also can decompose the polymer PDA to allow DOX release from the channels of mesoporous SiO$_2$. Most importantly, the photothermal conversion of GNR can also lead the liquid–gas phase transition of PFP to generate bubbles to accelerate the release of DOX, which can realize the chemotherapy of tumours. The subsequent synergistic chemo-PTT (contributed by the DOX and GNR) shows good anti-cancer activity. This work shows that the NIR-triggered multi-functional nanoplatform is of capital significance for future potential applications in drug delivery and cancer treatment.

# 1. Introduction

Global cancer morbidity and mortality is rising and will soon become the main cause of death [1]. So far, the currently available therapies that have been used clinically to treat metastatic cancer include aggressive surgery, radiotherapy and chemotherapy [2]. Among them, chemotherapy, a common method of tumour therapy in clinical application, relies on chemical drugs to kill cancer cells effectively. Despite being a mainstay in clinical cancer treatment, conventional chemotherapy still suffers from non-specific biodistribution, toxic side effects, tendency to produce drug resistance and less effective concentration [3–5]. With the continuous exploration of new methods to treat cancer in the medical field, photothermal therapy (PTT) based on nanomaterials has achieved excellent results [6].

Recently, PTT has been attracting considerable attention, which employs the new near-infrared (NIR) laser photo-absorbers to achieve laser thermal ablation of cancer cells [7]. In this process, once the tumour tissue is exposed to NIR light, the cell membrane structure and subcellular structure of cancer cells will be damaged due to the high temperature (approx. 45°C) generated by the laser photo-absorber, resulting in the destruction of the internal tissue structure of cancer cells and the loss of cellular activity [8,9]. Although being less toxic to healthy organs and tissues, PTT often fails to achieve the desired results when used alone [10]. Furthermore, due to uneven heat transfer and distribution of material within the tumour during PTT, incomplete destruction of the tumour and cancer cell recurrence usually appear. Therefore, a single nano-diagnostic system, which integrates PTT and chemotherapy, is expected to be a prospective research approach to improve overall efficacy.

At present, a large number of drug delivery systems have been widely reported, which can be applied to effective integrative collaborative treatment (chemotherapy + PTT), such as gold nanorods (GNR) coated with silica shell (GNR@MSNP) [11], gold nanoparticles modified hollow carbon system (DOX/HMC-Au@PEG) [12] and DTX/IR780 micelles [13]. Among them, Au-based nanomaterials can be used not only as an excellent photothermal conversion material, but also as a stable drug carrier. Therefore, Au-based nanomaterials have become excellent materials in chemo-photothermal diagnostic method. However, drug release rate is still limited to these drug delivery processes, even though pH and photothermal are both responsive to drug release [10,14]. Therefore, increasing the drug release rate is an urgent issue. However, designing the internal structure and composition of the drug carrier to directly promote the rapid release of the drug is considered a promising strategy.

Varieties of therapeutic agent-loaded bubbles were extensively used as ultrasound imaging contrast [15]. Because of the high image quality of bubbles, a therapeutic agent could achieve more precise ultrasound imaging and therapy simultaneously. Recently, ultrasound combined with bubbles mediates intracellular drug delivery via the interaction between cavitation of bubbles and cells [16]. Wang *et al*. prepared IR780/$Fe_3O_4$@PLGA/PFP/DOX nanoplatform. Under the photothermal effect, perfluoro-*n*-pentane (PFP) promotes the expansion of nanomaterials and the release of doxorubicin hydrochloride (DOX) [17].

In this study, we propose a NIR light-controlled combination therapy nanoplatform (GNR@m$SiO_2$-DOX/PFP@PDA) based on mesoporous silica-coated GNR, which is applied to synergistic chemo-PTT of tumours. In this nano-system, DOX and PFP were loaded into the pores of mesoporous $SiO_2$ simultaneously as a first step, then polydopamine (PDA) was grafted on the surface to act as a gatekeeper of the pores at neutral pH. PDA is highly sensitive to pH. So, drug molecules will be released at lower pH [18]. Previously, there were relatively few reports on multi-functional platforms that use bubbles to facilitate drug release. GNR undergoes photothermal conversion under the excitation of NIR light to generate a large amount of heat, resulting in a phase change of PFP. Subsequently, a large number of bubbles generated by the phase change of PFP promote the release of DOX in the m$SiO_2$ channels. Meanwhile, PDA is employed to encapsulate and carry DOX molecules at physiological temperature without premature release (scheme 1). Both the thermal effect of GNR and the bubbles generated by PFP weaken the electrostatic interaction force between drugs and m$SiO_2$. In the presence of PFP and NIR, the release amount of drugs is quite a lot. As the electrostatic effect diminishes, the DOX is disengaged from the multi-functional platform and released quickly. The thermal effect of GNR also continues the thermal treatment of tumours, resulting in combined PTT and highly effective chemotherapy for cancer treatment.

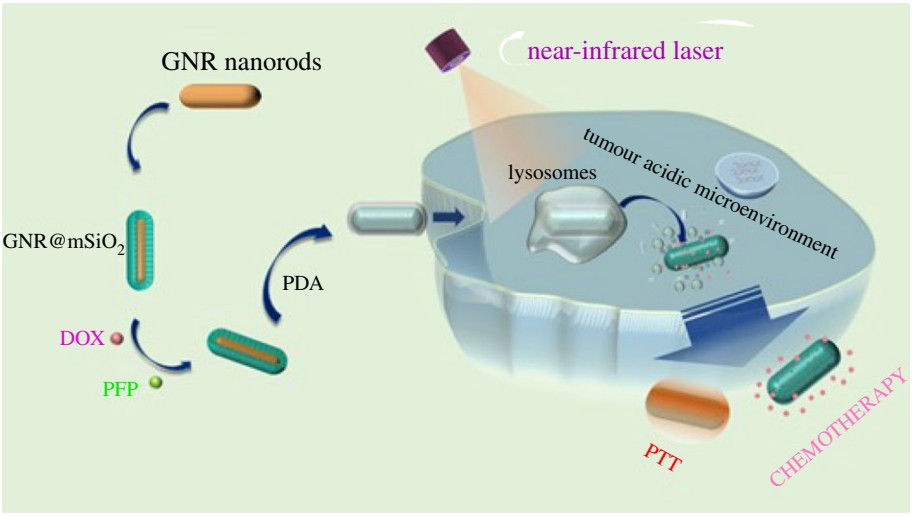

**Scheme 1.** Schematic illustration of phase change of nanometric therapeutic agents based on thermal effect of GNR for tumour chemotherapy and PTT.

# 2. Experimental section

## 2.1. Chemicals and materials

Chloroauric acid hydrate ($HAuCl_4 \cdot xH_2O$), DOX and sodium tetrahydroborate ($NaBH_4$) were obtained from Adamas. Cetyltrimethylammonium bromide (CTAB), silver nitrate ($AgNO_3$), L-ascorbic acid (AA), sodium hydroxide (NaOH) and tetraethyl orthosilicate (TEOS) were obtained from Sigma, Sinopharm Group Chemical Reagent Co. Ltd, General-Reagent, Alfa and Aldrich, respectively. Trimethylolaminomethane (Tris) was obtained from Sinopharm Group Chemical Reagent. PFP was purchased from Shanghai Fan Copula Technology Co. Ltd. Fetal bovine serum (FBS) and Dulbecco's modified Eagle's medium (DMEM) were purchased from GIBCO Life Technologies. CCK-8 Kit and Calcein-AM/PI (CA) were obtained from Dojindo Laboratories (Japan).

## 2.2. Characterization techniques

The microscopic morphology of nanomaterials was captured on a transmission electron microscope (TEM, Hitachi7800). We measured the absorption spectrum with an ultraviolet spectrophotometer (UVMini-1285). Fourier transform infrared spectroscopy (FTIR, Thermo Fisher Nicolet IS10) was used to characterize the spectra. Cell activity was measured by a Tecan microplate. The cell images were taken using an Olympus laser scanning confocal microscope (LSCM) imaging system.

## 2.3. Materials synthesis

### 2.3.1. Preparation of gold nanorods and GNR@mSiO₂

GNR was prepared by seed-mediated method according to published literature [19–21] and some improvements made on this basis. Typically, the first step is to prepare the seed solution. Specifically, 0.25 ml of 0.01 M $HAuCl_4 \cdot xH_2O$ was dispersed in 9.75 ml of 0.01 M CTAB, and then transferred into a 25 ml round-bottomed flask. After stirring gently for 2 min, 0.6 ml of 0.01 M $NaBH_4$ ($NaBH_4$ solution prepared with ice water and kept fresh) was added and kept stirring continuously for 10 min. After stirring, the solution was stood for 3 h at room temperature to obtain the seed solution. In addition, 10 ml of $HAuCl_4$ (0.01 M) was mixed with 190 ml of CTAB (0.1 M) together under magnetic stirring, followed by adding 2 ml of $AgNO_3$ (0.01 M) until mixed evenly. After adding 1.1 ml of L-AA, the colour of the solution changed from yellow to colourless. Then, 0.24 ml of as-prepared seed solution was added into the above mixture and kept slightly oscillating for 24 h at 30°C to prepare GNR.

The growth mechanism of mesoporous silica on the surface of nanoparticles mainly consists of three stages: silica oligomerizing process, the self-assembly process of silica/CTAB particles and subsequent agglomeration of silica/CTAB particles [22]. Owing to GNR mostly being surrounded by CTAB,

silica/CTAB particles will automatically aggregate on the surface of GNR, thus limiting the formation of free silica nanoparticles [22,23]. The concentration of CTAB in GNR solution plays a critical role in the formation of silica layer on GNR. Beyond a certain CTAB micelle concentration, no silica layer is formed on GNR surface [24]. Consequently, for the purpose of controlling the concentration of CTAB, GNR solution was washed twice and concentrated to 20 ml. Then, 200 µl of CTAB (0.1 M) was injected into a concentrated GNR solution to make the concentration of CTAB at 1 mM. The solution was stirred overnight to achieve equilibrium on the surface of GNR. The pH of the solution was controlled by dropping in 200 µl of NaOH solution, and then 400 µl of 20% TEOS–methanol mixture solution ($V_{TEOS}$: $V_{methanol}$) was injected and kept stirring for 24 h to get the final sample.

### 2.3.2. Preparation of GNR@mSiO$_2$-DOX

By mixing GNR@mSiO$_2$ (4 ml, 1.4 mg ml$^{-1}$) with DOX solution (500 µl, 1 mg ml$^{-1}$) and stirring continuously for 24 h, DOX can be easily loaded into the GNR@mSiO$_2$ mesoporous channels. After stirring, GNR@mSiO$_2$-DOX was obtained by centrifuging the mixed solution. Then, the DOX concentration in residual supernatant was determined from the ultraviolet absorption spectrum at 480 nm. The drug loading content and drug entrapment efficiency rate can be further estimated by the following calculation formulae, respectively:

$$w = \frac{m_{DOX}}{m},\tag{2.1}$$

where $w$ is drug loading content of DOX in the GNR@mSiO$_2$-DOX, $m_{DOX}$ is the weight of DOX loaded in GNR@mSiO$_2$-DOX and $m$ is the weight of GNR@mSiO$_2$-DOX, and

$$\mu = \frac{m_{DOX}}{M_{DOX}},\tag{2.2}$$

where $\mu$ is drug entrapment efficiency of DOX in the GNR@mSiO$_2$-DOX, $m_{DOX}$ is the weight of DOX loaded in GNR@mSiO$_2$-DOX and $M_{DOX}$ is the total weight of DOX before loading.

### 2.3.3. Preparation of PFP@GNR@mSiO$_2$-DOX

For the synthesis of GNR@mSiO$_2$-DOX/PFP, 5 ml of 1 mg ml$^{-1}$ GNR@mSiO$_2$-DOX NPs was transferred into a two-neck flask with rubber stopper, followed by evacuating for 3 min to remove air. Afterwards, 100 µl of PFP was injected into the flask and kept stirring for 5 min in an ice bath to obtain GNR@mSiO$_2$-DOX/PFP. After centrifugation, the precipitate was washed by ultrapure ice water several times to remove the excess PFP and stored at 4°C for further use.

### 2.3.4. Preparation of GNR@mSiO$_2$-DOX/PFP@PDA

To prevent premature leakage of the drug from the pores of the nanoplatform, a PDA film was coated on PFP@GNR@mSiO$_2$-DOX. Four milligrams PFP@GNR@mSiO$_2$-DOX and 2 mg dopamine hydrochloride solution were mixed in 4 ml Tris buffer (pH = 8.5) for 3 h in the dark with magnetic stirring. After centrifugation, the resultant PDA-coated GNR@mSiO$_2$-DOX/PFP particles were washed by ultrapure water several times and stored at 4°C for further use.

## 2.4. The photothermal effect promotes the formation of bubbles

The method to explore the photothermal conversion of GNR and GNR@mSiO$_2$ under NIR is as follows. First, the thermal effect of GNR is explored. Two millilitres of GNR nanoparticles were configured as 100 µg ml$^{-1}$, 200 µg ml$^{-1}$, 300 µg ml$^{-1}$, 400 µg ml$^{-1}$ and 500 µg ml$^{-1}$. They were exposed to NIR (808 nm laser, 2 W cm$^{-2}$) for 7 min. The temperature readings were saved every 15 s with a thermocouple probe. Second, the photothermal conversion of GNR@mSiO$_2$ nanomaterials with different concentrations (50, 100, 150 and 200 µg ml$^{-1}$) was explored. The detection method is consistent with that of gold nanoparticles.

As previous literature reported [25], the photothermal conversion efficiency ($\eta$) was calculated according to equation (2.3):

$$\eta = \frac{hM(t_{max} - t_{surr}) - Q_D}{I(1 - 10^{-A_{808}})},\tag{2.3}$$

where $h$ is heat transfer coefficient, $M$ is the surface area of the container, $t_{max}$ is the equilibrium temperature and $t_{surr}$ is the ambient surrounding temperature. $Q_D$ is heat lost to the surroundings, which was measured independently. And $A_{808}$ is the absorption intensity of GNR@mSiO$_2$ at 808 nm. The lumped quantity $hM$ is obtained on the basis of equation (2.4):

$$\tau_s = \frac{m_H C_H}{hM},\qquad(2.4)$$

where $\tau_s$ is the time constant for heat transfer, $m_H$ is the mass of ultrapure water and $C_H$ is heat capacity of ultrapure water. And $\tau_s$ is calculated by equation (2.5):

$$t = -\tau_s \ln \theta.\qquad(2.5)$$

$\tau_s$ can be calculated applying the time data from the cooling period versus negative natural logarithm of driving force temperature ($-\ln\theta$). And $\theta$ is calculated by equation (2.6):

$$\theta = \frac{t - t_{surr}}{t_{max} - t_{surr}}.\qquad(2.6)$$

To investigate the photothermal effect to promote the production of PFP bubbles, PFP@GNR@mSiO$_2$ solution was mixed into a 200 µg ml$^{-1}$ solution, and 20 µl of the solution was dropped onto a slide and covered with a coverslip. The solution was exposed to NIR (808 nm, 2 W cm$^{-2}$) for 5 min. The resulting bubbles were captured by an optical microscope after exposure to NIR irradiation.

## 2.5. The doxorubicin hydrochloride release experiment

In drug release experiments, the group of GNR@mSiO$_2$-DOX NPs was control; we could further explore the DOX release of GNR@mSiO$_2$-DOX/PFP NPs and to demonstrate that PFP can promote the release of DOX. First, GNR@mSiO$_2$-DOX/PFP NPs or GNR@mSiO$_2$-DOX NPs were prepared into an aqueous solution of 1 mg ml$^{-1}$. The drug loading efficiency of both was 3.1% by weight. The cumulative drug release was monitored when they were exposed to NIR (808 nm, 1 W cm$^{-2}$) for 0 h, 0.15 h. 0.5 h, 1 h, 2 h, 3 h, 4 h and 5 h, respectively. The ultraviolet absorption of DOX at 480 nm was used to calculate drug release.

## 2.6. *In vitro* cytotoxicity assay

Human cervical cancer cells (HeLa cells) were used to evaluate cytotoxicity *in vitro*. HeLa cells were incubated in DMEM containing 10% (v/v) FBS, penicillin (100 µg ml$^{-1}$) and streptomycin (100 µg ml$^{-1}$), and cultured in a 37°C atmosphere filled with 5% CO$_2$.

HeLa cell viability was determined using a CCK-8 (Cell Counting Kit-8). First, cells were inoculated into 96-well plates with 8000 cells per well and incubated for 24 h. And then, DMEM mixed with materials (GNR@mSiO$_2$, GNR@mSiO$_2$-DOX, GNR@mSiO$_2$-DOX/PFP and GNR@mSiO$_2$-DOX/PFP@PDA) was incubated with cells for 24 h, and each material was formulated as 0, 50, 100, 150 and 200 µg ml$^{-1}$, respectively. Cell viability was obtained based on the absorbance ratio of the control group. The absorbance was determined at 450 nm with a reference solution.

## 2.7. *In vitro* photothermal therapy effect and promote bubbles to accelerate doxorubicin hydrochloride release

First, HeLa cells were inoculated into a 96-well plate with 8000 cells per well, and DMEM mixed with materials (GNR@mSiO$_2$-DOX@PDA and GNR@mSiO$_2$-DOX/PFP@PDA) was incubated with cells for 4 h. Each material was formulated as 0, 50, 100, 150 and 200 µg ml$^{-1}$, respectively. The cells were exposed to an 808 nm NIR laser (2 W cm$^{-2}$) for 5 min and cultivated for 20 h to detect the cell viability.

For further observing the influence of PTT, Calcein-AM and PI co-staining was used. First of all, 12-well plates were used for inoculation of HeLa cells (150 000 cells per well) and incubation for 24 h. After washing with PBS, the cells were cultivated with GNR@mSiO$_2$-DOX@PDA and GNR@mSiO$_2$-DOX/PFP@PDA for 4 h, respectively. And then the cells exposed to NIR for 5 min (808 nm, 2 W cm$^{-2}$). They were incubated again for 4 h. Finally, they were incubated with Calcein-AM and PI for 15 min, and the images were observed with LSCM.

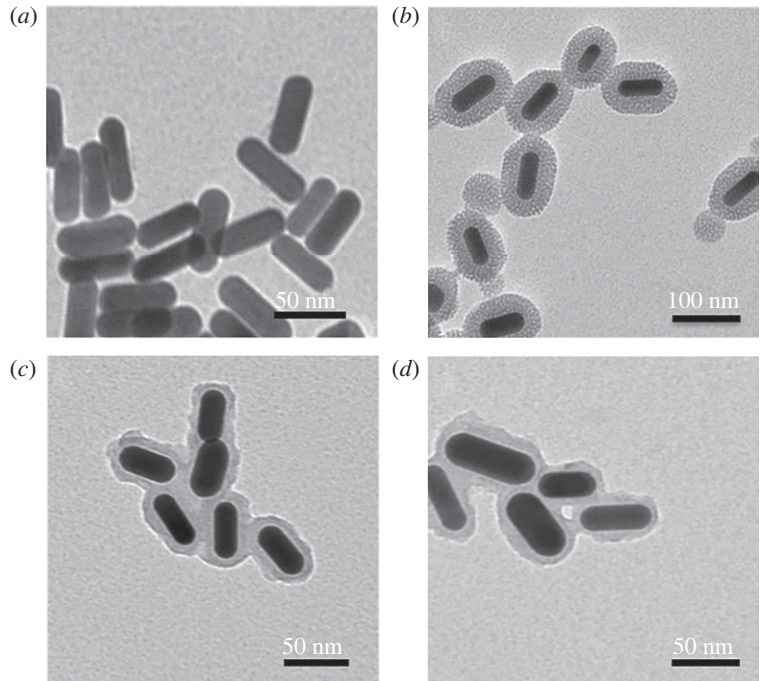

**Figure 1.** TEM images of (*a*) GNR, (*b*) GNR@mSiO$_2$, (*c*) GNR@mSiO$_2$-DOX@PDA and (*d*) GNR@mSiO$_2$-DOX/PFP@PDA.

## 2.8. *In vivo* photothermal imaging and photothermal therapy effect

Tumour-bearing mice were randomly divided into eight groups. In turn can be divided into control group, PBS + NIR, GNR@mSiO$_2$, GNR@mSiO$_2$-DOX@PDA, GNR@mSiO$_2$-DOX/PFP@PDA, GNR@mSiO$_2$ + NIR, GNR@mSiO$_2$-DOX@PDA + NIR and GNR@mSiO$_2$-DOX/PFP@PDA + NIR (*n* = 3). When the tumour is 5 mm in diameter, 100 µl of PBS, GNR@mSiO$_2$ (6 mg ml$^{-1}$), GNR@mSiO$_2$-DOX@PDA (6 mg ml$^{-1}$) or GNR@mSiO$_2$-DOX/PFP@PDA (6 mg ml$^{-1}$) wasere injected into the tumour of tumour-bearing mice within the corresponding group, respectively. Then, NIR irradiation was applied for 5 min (808 nm, 2 W cm$^{-2}$) 2 h after injection of the material, and the irradiation was performed once every 3 days for 14 days. During this process, the photothermal images were saved using infrared thermal imagers. Data on tumour volume and mouse weight were recorded at regular intervals. On the last day, the tumour and its vital organs were removed and haematoxylin staining (H&E) was performed.

# 3. Results and discussion

## 3.1. Characterization

In the process of mSiO$_2$ coating, CTAB layer is formed around the GNR, which is the organic template formed by mSiO$_2$. The TEM images show the successful preparation of GNR and GNR@mSiO$_2$. The homogeneous silica layer is about 20 nm thick and composed of disordered mesopores (figure 1*a*,*b*), which is naturally used as a drug carrier. The results of nitrogen adsorption/desorption isotherms show that the average pore diameter of mesoporous silica is about 2.19 nm, as shown in the electronic supplementary material, figure S1. The as-prepared GNR has a longitudinal surface plasmonic resonance (LSPR) peak at the NIR region (figure 2*a*). After being wrapped with silica shell, GNR@mSiO$_2$ exhibits a small red-shift (approx. 22 nm), which also verifies the corresponding theoretical simulation [26]. Because of the large distance between the GNR in GNR@mSiO$_2$, the agglomeration phenomenon does not affect the position of the SPR peak in the NIR window [26]. Therefore, photons are allowed to penetrate biological tissues with a high transmittance, which further indicates that GNR@mSiO$_2$ can be naturally applied in epithelial tissues [27,28]. The negative potential of GNR@mSiO$_2$ and the positive potential of DOX provide another theoretical basis for GNR@mSiO$_2$ loaded with DOX (figure 2*b*). Anti-cancer drugs are delivered under the activation of NIR light, and precise control of drug area, time and dose can be achieved by precise control of light [29]. The

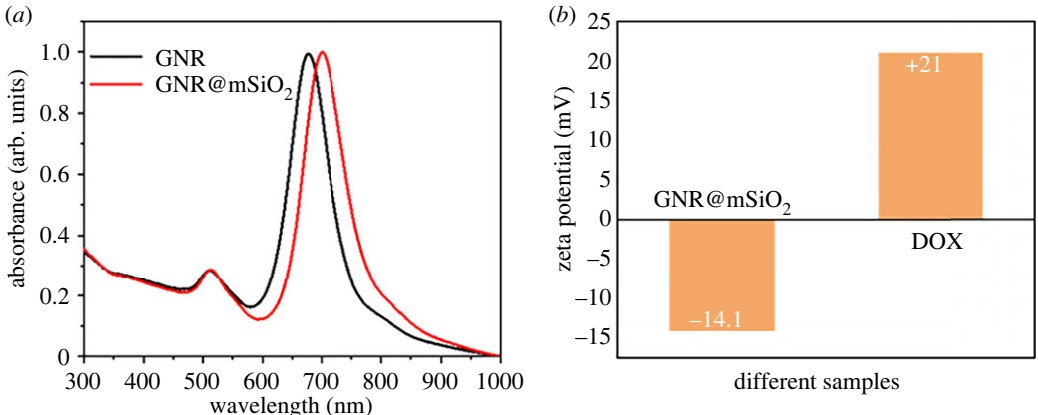

**Figure 2.** (a) UV–visible curves of GNR and GNR@mSiO$_2$. (b) Surface potential of GNR@mSiO$_2$ and DOX.

release amount of DOX is calculated according to DOX-fitting standard curve (electronic supplementary material, figure S2).

FTIR technology was used to verify the successful synthesis of PFP@GNR@mSiO$_2$ modified by PDA shell. As shown in the electronic supplementary material, figure S3, the absorption peaks at 1500 cm$^{-1}$ indicated the overlap of the C=C resonance vibrations and the N–H bending vibrations of PDA [30,31]. The rough surface of nanoparticles can be seen in TEM images of GNR@mSiO$_2$-DOX@PDA and GNR@mSiO$_2$-DOX/PFP@PDA from figure 1c,d. Both of them indicate that PDA was successfully coated on the mesoporous silica surface.

## 3.2. The photothermal conversion and drug release

GNR can exhibit excellent properties when the laser wavelength matches its LSPR peak, which effectively converts photoenergy into heat characterized by UV spectroscopy. GNR and GNR@mSiO$_2$ show strong absorption in the NIR range, and better photothermal effect can be achieved by NIR excitation. In this experiment, the photothermal properties of GNR and GNR@mSiO$_2$ were evaluated. The photothermal conversion effect of GNR and GNR@mSiO$_2$ under NIR continuous wavelength laser irradiation was evaluated by thermocouple. The temperature of GNR and GNR@mSiO$_2$ solutions increased with increasing concentration (50, 100, 150 and 200 µg ml$^{-1}$) at a certain power. The photothermal effect of GNR-coated mesoporous silica is almost not weakened (figure 3a,b). The temperature of GNR@mSiO$_2$ solution varied from 20°C to 27°C after 7 min of laser irradiation at 808 nm (2 W cm$^{-2}$). It can be concluded that GNR still shows good photothermal effect after silica coating. In order to deeply study the photothermal effect of GNR@mSiO$_2$, we observed the temperature change of GNR@mSiO$_2$ exposed to irradiation of 808 nm and made the time fitting according to a previously reported method [32,33]. As shown in figure 3c,d, time constant for heat transfer from the system was obtained to be $\tau_s = 363$ s, which applied the linear time data of the cooling period (after 840 s) versus negative natural logarithm of driving force temperature. Substituting the value of $\tau_s$ into equation (2.4), $m_H$ is 2 g and the $C_H$ is 4.2 J g$^{-1}$, $hM$ can be obtained. And the value of $hM$ is substituted into equation (2.3). The photothermal conversion efficiency ($\eta$) of GNR@mSiO$_2$ reached 44.85%. The photothermal efficiency of GNR@mSiO$_2$ nanomaterials is much higher than that of other materials used as PPT carriers, such as Cu$_{2-x}$Se nanocrystals (22%) and Cu$_9$S$_5$ nanocrystals (25.7%) [25,34]. The boiling point of PFP is 29°C. When blood pressure rises after intravenous injection, the boiling point of PFP rises to 40–50°C [35,36]. These results show that GNR@mSiO$_2$ has a photothermal effect when exposed to NIR and can reach the phase transition temperature of PFP to promote the release of DOX.

To further study additional uses of GNR@mSiO$_2$-DOX/PFP NPs to be drug carriers for tumour therapy, bubble-triggered drug release attracted our attention. First, the thermal effect promoting bubble generation exposed to NIR (2 W cm$^{-2}$, 5 min), we used a microscope to collect images of GNR@mSiO$_2$-PFP solutions with different concentrations on glass slides. Before laser irradiation, there is basically no bubble generation (figure 4a). After NIR irradiation, it was observed through the microscope that a large number of bubbles were generated in GNR@mSiO$_2$-PFP solution (figure 4b) and gradually increased. These results indicate that after laser irradiation, the temperature of the solution reaches the gasification threshold of phase change agent, phase change occurs in PFP, and bubbles are released from GNR@mSiO$_2$-PFP. At a power density of 2 W cm$^{-2}$, the number of bubbles

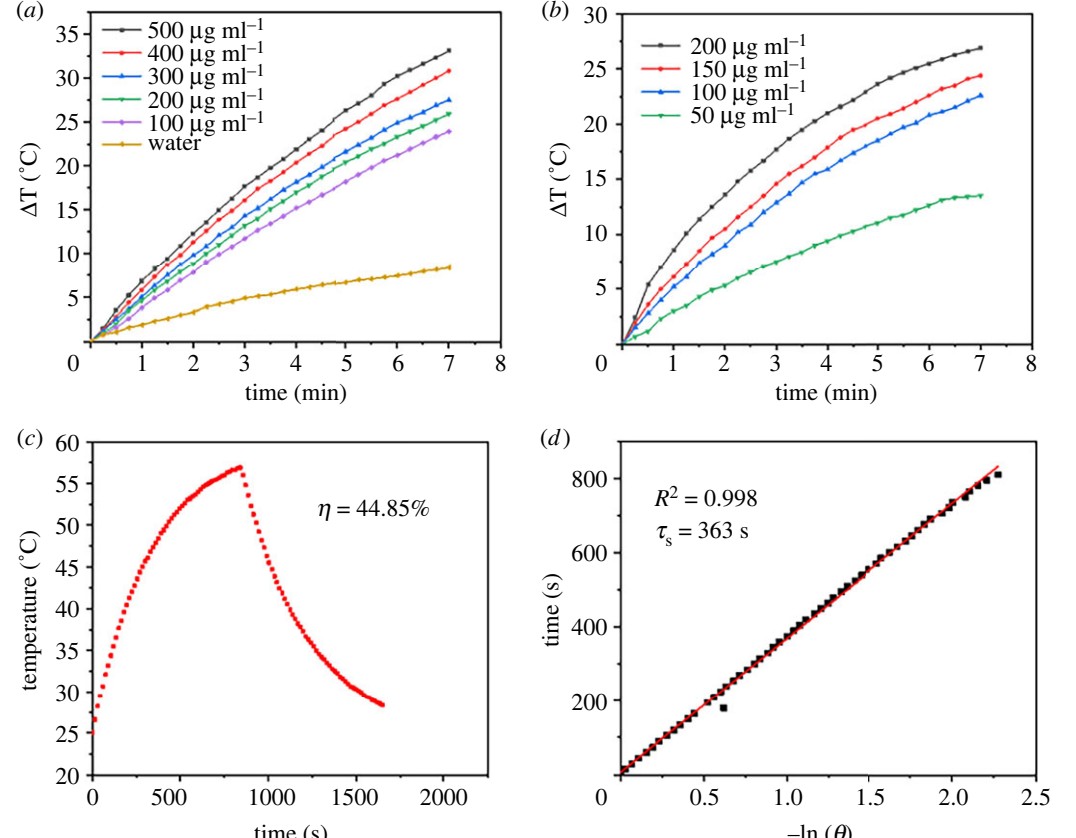

**Figure 3.** (a) Temperature elevation curves of GNR solutions with different concentrations under 808 nm laser irradiation at a power density of 2 W cm$^{-2}$ for 7 min. (b) Temperature elevation curves of GNR@mSiO$_2$ solutions with different concentrations under 808 nm laser irradiation at a power density of 2 W cm$^{-2}$ for 7 min. (c) The monitored temperature changes over 1600 s with GNR@SiO$_2$ (200 µg ml$^{-1}$) after 808 nm laser irradiation (2 W cm$^{-2}$) for 840 s, following which the laser was turned off. (d) The calculation of the time constant for heat transfer from the system using linear regression of the cooling profile.

increases with increasing concentration of GNR@mSiO$_2$-PFP solution. Under the laser irradiation of GNR@mSiO$_2$-PFP solutions with different concentrations, the number of bubbles generated increases with the continuous increase of GNR@mSiO$_2$-PFP solution concentration. The number of bubbles produced was quantified, and figure 4c shows that the number of bubbles increased with the increase of concentration under a certain power. Electronic supplementary material, figure S4, shows that the number of bubbles increases with increasing concentration. Therefore, conditions are provided for bubbles to accelerate DOX release. Based on the above results, it was further substantiated that PFP accelerated drug release under the thermal effect of GNR.

To investigate the DOX release characteristics in GNR@mSiO$_2$-DOX/PFP, we also selected GNR@mSiO$_2$-DOX without PFP as the control. Both materials will have DOX release under laser irradiation (figure 4d). Under excitation of 808 nm, the amount of DOX released by GNR@mSiO$_2$-DOX/PFP (43.6%) is far superior to the amount released by GNR@mSiO$_2$-DOX (20.9%) in the same release time. The dose released in the environment was quantitatively analysed by an ultraviolet spectrophotometer. The release rate of GNR@mSiO$_2$-DOX/PFP in a short time under laser irradiation reached nearly 50%. It has been reported that the release of DOX in GNR@mSiO$_2$-DOX system under laser irradiation can only reach 28% within 5 h under acidic environment [26]. Therefore, GNR@mSiO$_2$-DOX/PFP is an excellent drug delivery system, which can effectively shorten the drug release time and has a broad application prospect. The acid- and photo-triggered drug release was also further investigated. Electronic supplementary material, figure S5, shows that the GNR@mSiO$_2$-DOX/PFP NPs with NIR released more than 70% DOX at pH 5.7 compared with no irradiation.

## 3.3. *In vitro* anti-tumour activity

The cytotoxicity of GNR@mSiO$_2$-DOX/PFP to HeLa cells was determined via a Cell Counting Kit-8. As can be seen from figure 5, cells were cultivated with GNR@mSiO$_2$ NPs and GNR@mSiO$_2$-DOX NPs, and

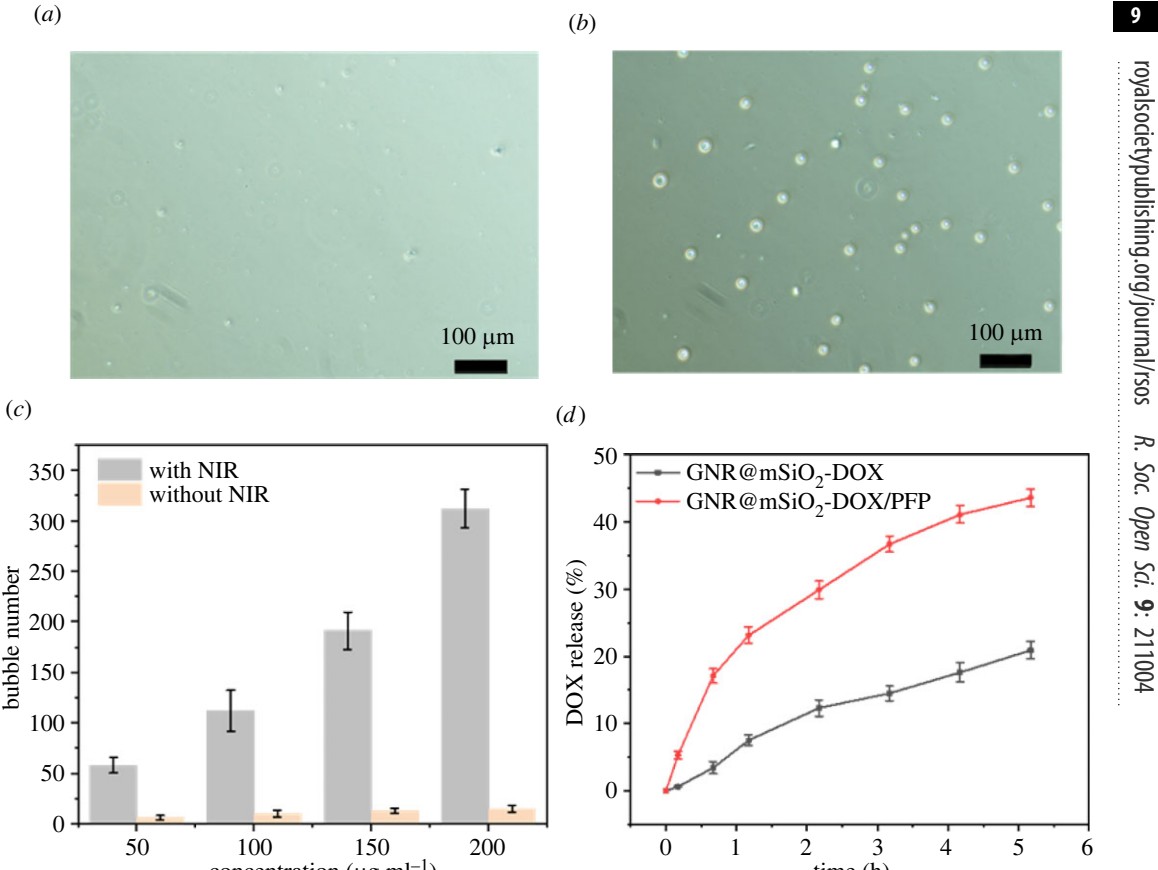

**Figure 4.** Microscope images of microbubbles from GNR@mSiO$_2$-PFP (*a*) before NIR irradiation and (*b*) after NIR irradiation for 5 min (2 W cm$^{-2}$; scale bar = 100 μm). (*c*) Number of microbubbles in PFP@GNR@mSiO$_2$ solutions with NIR laser irradiation (power density of 2 W cm$^{-2}$) for 5 min or without NIR. (*d*) DOX release profiles from GNR@SiO$_2$-DOX and GNR@SiO$_2$-DOX/PFP with NIR laser irradiation (power density of 2 W cm$^{-2}$).

there was no significant change in cell viability. With increasing concentration of them, the cell viabilities showed no obvious decrease, indicating that they had good biocompatibility. However, with increasing concentrations of GNR@mSiO$_2$-DOX/PFP NPs, cell viabilities decreased gradually without NIR, demonstrating that GNR@mSiO$_2$-DOX/PFP NPs were cytotoxic. While PFP was essentially non-toxic [37]. It is indicated that partial PFP phase transition may occur to promote the release of a small part of DOX, thus leading to the decrease of cell viability. PDA is biocompatible and degrades in a slightly acidic environment, so it is the best polymer to prevent DOX leakage. PDA is a promising molecule that can act as an adhesive layer, which can be used to encapsulate DOX molecules and store them at neutral pH. PDA films will be disintegrated in an acidic environment. As shown in figure 5*a*, GNR@mSiO$_2$-DOX/PFP coated with PDA increased the cell survival rate to over 85%. Subsequently, the chemotherapy cytotoxicities of GNR@mSiO$_2$-DOX@PDA and GNR@mSiO$_2$-DOX/PFP@PDA on cells were investigated under NIR. As shown in figure 5*b*, when irradiated for 5 min, the cell viability of GNR@mSiO$_2$-DOX/PFP@PDA decreased compared to that of GNR@mSiO$_2$-DOX@PDA, demonstrating that PFP effectively promoted DOX release under the photothermal effect of GNR. Electronic supplementary material, figure S6, shows that the cell viability of BALB/3T3 cells exposed to NIR (power density of 2 W cm$^{-2}$) for 5 min was almost the same as that of unirradiated cells. Calcein-AM/PI double staining with CLSM was performed to further observe the killing effect of GNR@mSiO$_2$-DOX/PFP@PDA on cells. Green (Calcein-AM) fluorescence represents living cells, and red (PI) fluorescence represents dead or late-apoptotic cells. Cells incubated with GNR@mSiO$_2$-DOX/PFP@PDA or GNR@mSiO$_2$ exposed to an 808 nm laser, and the signal of GNR@mSiO$_2$-DOX/PFP@PDA group was almost always red fluorescent signal (figure 6), which indicated that GNR@mSiO$_2$-DOX/PFP@PDA can effectively kill cancer cells.

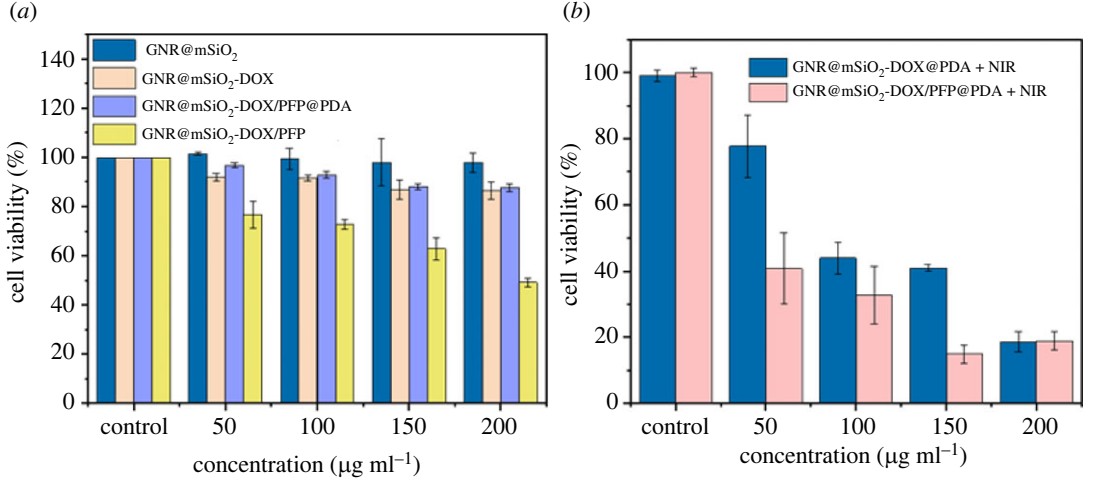

**Figure 5.** (*a*) Cell cytotoxicity of GNR@mSiO$_2$, GNR@mSiO$_2$-DOX, GNR@mSiO$_2$-DOX/PFP@PDA and GNR@mSiO$_2$-DOX/PFP to HeLa cells incubated for 24 h. (*b*) *In vitro* PPT effect and chemotherapy of GNR@mSiO$_2$-DOX@PDA and GNR@mSiO$_2$-DOX/PFP@PDA in the presence or absence of 808 nm NIR.

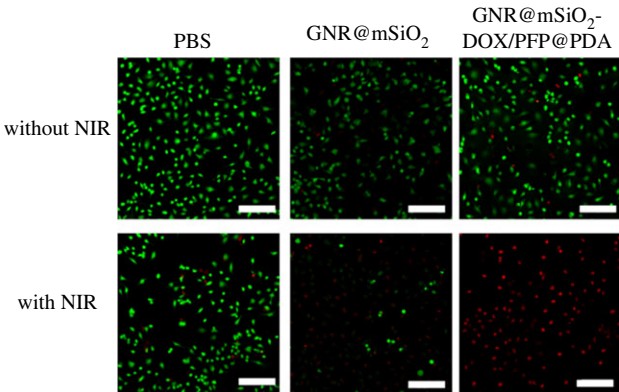

**Figure 6.** Confocal images of Calcein-AM/PI-stained HeLa cells after incubation with GNR@mSiO$_2$ and GNR@mSiO$_2$-DOX/PFP@PDA and irradiation with 808 nm laser at 2 W cm$^{-2}$ for 5 min (scale bar = 150 μm).

## 3.4. Photothermal imaging and photothermal therapy effect *in vivo*

A HeLa tumour-bearing mouse model was adopted to explore the influence of infrared thermal imaging *in vivo*, and the combination therapy of chemotherapy and PTT. HeLa tumour-bearing mice were randomly divided into eight groups: PBS, PBS + NIR, GNR@mSiO$_2$, GNR@mSiO$_2$ + NIR, GNR@mSiO$_2$-DOX@PDA, GNR@mSiO$_2$-DOX@PDA + NIR, GNR@mSiO$_2$-DOX/PFP@PDA and GNR@mSiO$_2$-DOX/PFP@PDA + NIR. Figure 7 shows the infrared thermal images of GNR@mSiO$_2$-DOX/PFP@PDA NPs *in vivo*. After injecting PBS and the other several kinds of materials into the tumour, there was a wait of 2 h. Tumours of mice injected with saline had no significant temperature rise. By contrast, for mice injected with NPs, the temperature of tumours rapidly increased to 50°C under NIR laser irradiation. Weight changes and tumour volume changes in each mouse were recorded every other day during the 14-day treatment, and the results were expressed as the mean ± s.d. The weight of each group of mice changed so little during the 14 days that there was no statistical significance (figure 8*a*). Both the control and experimental mice were healthy after laser irradiation, indicating that the laser did no harm to the health of the mice. We recorded the relative tumour volume changes in mice, as shown in figure 8*b*. The tumour volumes of the mice in groups (PBS, GNR@mSiO$_2$, GNR@mSiO$_2$-DOX@PDA, GNR@mSiO$_2$-DOX/PFP@PDA) apparently increase over time, and the tumour volume was about 4.5 times larger than the initial value, indicating that NIR irradiation alone and the nano-system (GNR@mSiO$_2$, GNR@mSiO$_2$-DOX@PDA and GNR@mSiO$_2$-DOX/PFP@PDA) without NIR irradiation did not have effective prohibition ability on tumour growth. In contrast, the tumour growth in groups (GNR@mSiO$_2$ + NIR, GNR@mSiO$_2$-DOX@PDA +

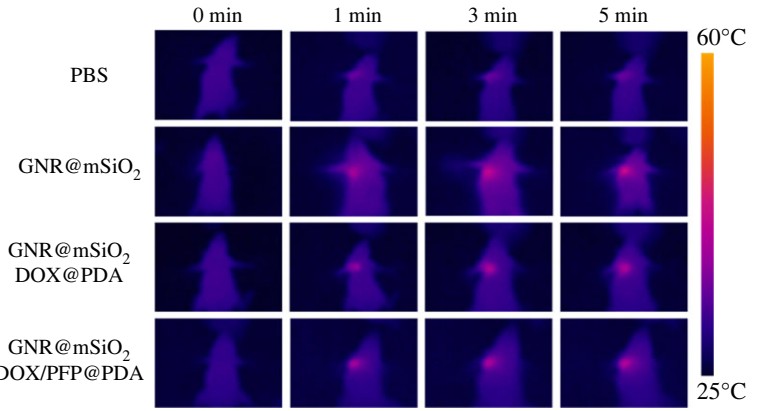

**Figure 7.** IR thermal images of tumour-bearing mice following intratumoral injection of PBS, GNR@mSiO$_2$, GNR@mSiO$_2$-DOX@PDA and GNR@mSiO$_2$-DOX/PFP@PDA solutions under 808 nm laser irradiation at a power density of 2 W cm$^{-2}$ for 5 min.

**Figure 8.** (*a*) Body weight of mice after various treatments. (*b*) Tumour growth curves of mice from different groups after treatment (mean ± s.d., ***$p < 0.001$). (*c*) Pictures of tumours at the end of 14-day treatments.

NIR and GNR@mSiO$_2$-DOX/PFP@PDA + NIR) is greatly inhibited. Compared with the control group (PBS), the tumour inhibitory ratio in GNR@mSiO$_2$-DOX/PFP@PDA + NIR group is estimated to be about 79%. Comparisons among the groups were evaluated by *t*-test (two variables) or ANOVA (multiple variables) followed by the Student–Newman–Keuls test. A value of $p < 0.001$ was obtained,

which was considered statistically significant. The results demonstrate that the group of GNR@mSiO$_2$-DOX/PFP@PDA + NIR accelerated and promoted the release of drugs because of the PTT effect, and then combined with the photothermal treatment, the therapeutic effect on the tumour was more obvious. After 14 days of treatment, the mice were dissected and the tumours were excised. The typical photographs of excised tumours (figure 8c) also indicate that the treatment of GNR@mSiO$_2$-DOX/PFP@PDA with 808 nm irradiation dramatically inhibits the tumour growth, and the treated mouse has the lowest tumour size. Furthermore, there were no remarkable pathological lesions or harm to the tumour tissue in the PBS group or the experiment without NIR (electronic supplementary material, figure S7a). Compared with the experimental plus NIR, the tumour tissue damage was obvious and necrotic areas appeared. Besides, H&E staining of vital organs showed no damage to the solid heart, liver, spleen, lungs and kidneys (electronic supplementary material, figure S7b).

## 4. Conclusion

In summary, we have successfully developed a multi-functional nanoplatform based on the photothermal effect of GNR and bubble-accelerated DOX release for combined chemotherapy and PTT of tumours. The nanoplatform GNR@mSiO$_2$-DOX/PFP@PDA has good biocompatibility, strong NIR photothermal conversation efficiency, controllable drug release acceleration and synergistic chemo-photothermal tumour therapy capability. Under the excitation of an 808 nm NIR laser, the thermal generation of GNR in the nanoplatform GNR@mSiO$_2$-DOX/PFP@PDA can decompose the polymer as well as further induce the liquid–gas phase transformation of PFP to realize acceleration of the drug release. This nanoplatform is not cytotoxic because it can only release the drug inside and leads to superior synergistic chemo-PTT of tumours under NIR light irradiation. It is expected that the design of DOX-gated nanoplatform provides an efficient strategy for the preparation of stable and safe drug delivery systems for tumour treatment.

Ethics. All animal experiments were conducted in accordance with the requirements of the laws of the People's Republic of China (GB14925-2010) and were approved by the Shanghai Municipal Science and Technology Commission (SYXK (Shanghai) 2019-0020).

Data accessibility. The data are provided in the electronic supplementary material [38].

Authors' contributions. C.F.: data curation, investigation, methodology and writing—original draft; J.L.: data curation, investigation and writing—original draft; Y.W.: funding acquisition, methodology and writing—review and editing; Y.L.: data curation and investigation; J.L.: conceptualization, funding acquisition, methodology, resources, supervision and writing—review and editing.

All authors gave final approval for publication and agreed to be held accountable for the work performed therein.

Conflict of interest declaration. The authors declare that they have no known competing financial interests or personal relationships that could have appeared to influence the work reported in this paper.

Funding. We acknowledge the financial support from National Natural Science Foundation of China (grant nos. 11905123 and 81971740) and Innovative Research Team of High-Level Local Universities in Shanghai.

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
