## [Peer Review File · Royal Society Open Science]

Review History

RSOS-211004.R0 (Original submission)

Review form: Reviewer 1

Is the manuscript scientifically sound in its present form?

No

Are the interpretations and conclusions justified by the results?

No

Is the language acceptable?

No

Do you have any ethical concerns with this paper?

No

Have you any concerns about statistical analyses in this paper?

No

Recommendation?

Reject

Comments to the Author(s)

In this manuscript, the authors developed a multifunctional nanoplatform (GNR@mSiO₂-DOX/PFP@PDA) based on gold nanorods (GNR) coated with mesoporous silica (mSiO₂) for synergistic chemo-photothermal therapy of tumors. Doxorubicin hydrochloride (DOX) and perfluoro-n-pentane (PFP) were loaded into the pores of mSiO₂. A polydopamine (PDA) layer as the gatekeeper of pores was coated on their surface. The whole nano-system is too complex, and the novelty of this work is very poor. There are a lot of similar published works, this manuscript is not suitable for publication on Royal Society Open Science.

1. What is the function of PFP in this nano-system?
2. In Figure 3a and b, one cannot see the difference within different concentrations of GNR solutions and GNR@mSiO₂ solutions. These experiments should redo.
3. What is the purpose of microbubbles generation of GNR@mSiO₂-PFP?
4. The error bars are missing in Figure 4d.
5. The in vivo chemo-photothermal therapeutic effect is very poor, but the authors claimed that 'the tumor could be basically completely eliminated after 8 ~ 10 days (Figure 8b).' The results are inconsistent with Figure 8c.

Review form: Reviewer 2**Is the manuscript scientifically sound in its present form?**

Yes

Are the interpretations and conclusions justified by the results?

Yes

Is the language acceptable?

Yes

Do you have any ethical concerns with this paper?

No

Have you any concerns about statistical analyses in this paper?

Yes

Recommendation?

Accept with minor revision (please list in comments)

Comments to the Author(s)

The present study provides a NIR light controlled multifunctional nanoplatform based on gold nanorods coated with mesoporous silica for synergistic chemo-photothermal therapy of tumors. The manuscript is well managed and the work is interesting. However, further work should be done and the paper should be minor revised according to the following points before its acceptance.

1. The authors claimed that the photothermal conversion of GNR can also lead the liquid-gas phase transition of PFP to generate the bubbles to accelerate the release of DOX. For this reason, the control group without NIR in Figure 4c should be added.

2. To evaluate the in vitro cellular cytotoxicity, it is suggested that the cell viability in normal cells should also be investigated, and the irradiation power of NIR in Figure 5b should also be mentioned.
3. For in vivo study of tumor growth curves in figure 8d, statistical analysis should be added.
4. The manuscript contains some grammar errors and inconsistent formats in references section, please check them carefully.
5. There are some writing mistakes which should be carefully examined and revised. For example, in Figure 3d "R2 = 0.998" should be "R2 =0.998".

Decision letter (RSOS-211004.R0)

Dear Dr Liu:

Title: Chemodrug-gated mesoporous nanoplatform for NIR light controlled drug release and synergistic chemophothermal therapy of tumors
Manuscript ID: RSOS-211004

The editor assigned to your manuscript has now received comments from reviewers. We would like you to revise your paper in accordance with the referee and Subject Editor suggestions which can be found below (not including confidential reports to the Editor). Please note this decision does not guarantee eventual acceptance.

Please submit your revised paper before 10-Dec-2021. Please note that the revision deadline will expire at 00.00am on this date. If we do not hear from you within this time then it will be assumed that the paper has been withdrawn. In exceptional circumstances, extensions may be possible if agreed with the Editorial Office in advance. We do not allow multiple rounds of revision so we urge you to make every effort to fully address all of the comments at this stage. If deemed necessary by the Editors, your manuscript will be sent back to one or more of the original reviewers for assessment. If the original reviewers are not available we may invite new reviewers.

Please also include the following statements alongside the other end statements. As we cannot publish your manuscript without these end statements included, if you feel that a given heading is not relevant to your paper, please nevertheless include the heading and explicitly state that it is not relevant to your work.

- Ethics statement

Please clarify whether you received ethical approval from a local ethics committee to carry out your study. If so please include details of this, including the name of the committee that gave consent in a Research Ethics section after your main text. Please also clarify whether you received informed consent for the participants to participate in the study and state this in your Research Ethics section.

OR

Please clarify whether you obtained the necessary licences and approvals from your institutional animal ethics committee before conducting your research. Please provide details of these licences and approvals in an Animal Ethics section after your main text.

OR

Please clarify whether you obtained the appropriate permissions and licences to conduct the fieldwork detailed in your study. Please provide details of these in your methods section.

- Data accessibility

It is a condition of publication that you make available the data and research materials supporting the results in the article. Datasets should be deposited in an appropriate publicly available repository and details of the associated accession number, link or DOI to the datasets must be included in the Data Accessibility section of the article (<https://royalsocietypublishing.org/rsos/for-authors#question17>). Reference(s) to datasets should also be included in the reference list of the article with DOIs (where available).

Please include a Data Availability section after your main text stating where supporting data are available from, or where they will be made available should your article be accepted for publication.

If you wish to submit your supporting data or code to Dryad (<http://datadryad.org/>), or modify your current submission to dryad, please use the following link:
<http://datadryad.org/submit?journalID=RSOS&manu=RSOS-211004>

- Competing interests

Please include a Competing Interests section after your main text declaring any financial or non-financial competing interests. If you have no competing interests please state 'I/we have no competing interests.'

- Authors' contributions

Please include an Authors' Contributions section at the end of your main text detailing the contribution of each author. All authors should have read and approved the manuscript before submission and this should be stated in the Authors' Contributions section.

The list of Authors should meet all of the following criteria; 1) substantial contributions to conception and design, or acquisition of data, or analysis and interpretation of data; 2) drafting the article or revising it critically for important intellectual content; and 3) final approval of the version to be published.

- Acknowledgements

- Funding statement

Please include a funding section after your main text which lists the source of funding for each author.

Yours sincerely,
Dr Ellis Wilde
Publishing Editor, Journals

On behalf of the Subject Editor Professor Anthony Stace and the Associate Editor Dr Dattatray Late.

RSC Associate Editor
Comments to the Author:
Major Revision

RSC Subject Editor
Comments to the Author:
(There are no comments.)

Reviewers' Comments to Author:

Reviewer: 1

Comments to the Author(s)

In this manuscript, the authors developed a multifunctional nanoplatform (GNR@mSiO₂-DOX/PFP@PDA) based on gold nanorods (GNR) coated with mesoporous silica (mSiO₂) for synergistic chemo-photothermal therapy of tumors. Doxorubicin hydrochloride (DOX) and perfluoro-n-pentane (PFP) were loaded into the pores of mSiO₂. A polydopamine (PDA) layer as the gatekeeper of pores was coated on their surface. The whole nano-system is too complex, and the novelty of this work is very poor. There are a lot of similar published works, this manuscript is not suitable for publication on Royal Society Open Science.

1. What is the function of PFP in this nano-system?

2. In Figure 3a and b, one cannot see the difference within different concentrations of GNR solutions and GNR@mSiO₂ solutions. These experiments should redo.
3. What is the purpose of microbubbles generation of GNR@mSiO₂-PPF?
4. The error bars are missing in Figure 4d.
5. The in vivo chemo-photothermal therapeutic effect is very poor, but the authors claimed that 'the tumor could be basically completely eliminated after 8 ~ 10 days (Figure 8b).' The results are inconsistent with Figure 8c.

Reviewer: 2

Comments to the Author(s)

The present study provides a NIR light controlled multifunctional nanoplatform based on gold nanorods coated with mesoporous silica for synergistic chemo-photothermal therapy of tumors. The manuscript is well managed and the work is interesting. However, further work should be done and the paper should be minor revised according to the following points before its acceptance.

1. The authors claimed that the photothermal conversion of GNR can also lead the liquid-gas phase transition of PFP to generate the bubbles to accelerate the release of DOX. For this reason, the control group without NIR in Figure 4c should be added.
2. To evaluate the in vitro cellular cytotoxicity, it is suggested that the cell viability in normal cells should also be investigated, and the irradiation power of NIR in Figure 5b should also be mentioned.
3. For in vivo study of tumor growth curves in figure 8d, statistical analysis should be added.
4. The manuscript contains some grammar errors and inconsistent formats in references section, please check them carefully.
5. There are some writing mistakes which should be carefully examined and revised. For example, in Figure 3d "R2 = 0.998" should be "R2 =0.998".

Author's Response to Decision Letter for (RSOS-211004.R0)

See Appendix A.

RSOS-211004.R1 (Revision)

Review form: Reviewer 1

Is the manuscript scientifically sound in its present form?

Yes

Are the interpretations and conclusions justified by the results?

Yes

Is the language acceptable?

Yes

Do you have any ethical concerns with this paper?

No

Have you any concerns about statistical analyses in this paper?

Yes

Recommendation?

Accept as is

Comments to the Author(s)

The manuscript has been revised mostly according to the comments and suggestions of the reviewers, I recommend the acceptance of the manuscript for publication in Royal Society Open Science.

Review form: Reviewer 2

Is the manuscript scientifically sound in its present form?

Yes

Are the interpretations and conclusions justified by the results?

Yes

Is the language acceptable?

Yes

Do you have any ethical concerns with this paper?

No

Have you any concerns about statistical analyses in this paper?

Yes

Recommendation?

Accept with minor revision (please list in comments)

Comments to the Author(s)

The authors have addressed all my questions except the statistical analysis in figure 8b. After adding the statistical analysis, this work could be published in Royal Society Open Science.

Decision letter (RSOS-211004.R1)

Dear Dr Liu:

Title: Chemodrug-gated mesoporous nanoplatform for NIR light controlled drug release and synergistic chemophotothermal therapy of tumors
Manuscript ID: RSOS-211004.R1

Thank you for submitting the above manuscript to Royal Society Open Science. On behalf of the Editors and the Royal Society of Chemistry, I am pleased to inform you that your manuscript will be accepted for publication in Royal Society Open Science subject to minor revision in accordance with the referee suggestions. Please find the reviewers' comments at the end of this email.

The reviewers and handling editors have recommended publication, but also suggest some minor revisions to your manuscript. Therefore, I invite you to respond to the comments and revise your manuscript.

Please also include the following statements alongside the other end statements. As we cannot publish your manuscript without these end statements included, if you feel that a given heading is not relevant to your paper, please nevertheless include the heading and explicitly state that it is not relevant to your work. We have included a screenshot example of the end statements for reference.

- Ethics statement

Please clarify whether you received ethical approval from a local ethics committee to carry out your study. If so please include details of this, including the name of the committee that gave consent in a Research Ethics section after your main text. Please also clarify whether you received informed consent for the participants to participate in the study and state this in your Research Ethics section.

OR

Please clarify whether you obtained the necessary licences and approvals from your institutional animal ethics committee before conducting your research. Please provide details of these licences and approvals in an Animal Ethics section after your main text.

OR

Please clarify whether you obtained the appropriate permissions and licences to conduct the fieldwork detailed in your study. Please provide details of these in your methods section.

- Data accessibility

It is a condition of publication that you make available the data and research materials supporting the results in the article. Datasets should be deposited in an appropriate publicly available repository and details of the associated accession number, link or DOI to the datasets must be included in the Data Accessibility section of the article (<https://royalsocietypublishing.org/rsos/for-authors#question17>). Reference(s) to datasets should also be included in the reference list of the article with DOIs (where available).

Please include a Data Availability section after your main text stating where supporting data are available from, or where they will be made available should your article be accepted for publication.

If you wish to submit your supporting data or code to Dryad (<http://datadryad.org/>), or modify your current submission to dryad, please use the following link:
<http://datadryad.org/submit?journalID=RSOS&manu=RSOS-211004.R1>

- Competing interests

Please include a Competing Interests section after your main text declaring any financial or non-financial competing interests. If you have no competing interests please state 'I/we have no competing interests.'

- Authors' contributions

Please include an Authors' Contributions section at the end of your main text detailing the contribution of each author. All authors should have read and approved the manuscript before submission and this should be stated in the Authors' Contributions section.

The list of Authors should meet all of the following criteria; 1) substantial contributions to conception and design, or acquisition of data, or analysis and interpretation of data; 2) drafting the article or revising it critically for important intellectual content; and 3) final approval of the version to be published.

- Acknowledgements

- Funding statement

Please include a funding section after your main text which lists the source of funding for each author.

Because the schedule for publication is very tight, it is a condition of publication that you submit the revised version of your manuscript before 19-Feb-2022. Please note that the revision deadline will expire at 00.00am on this date. If you do not think you will be able to meet this date please let me know immediately.

- 1) A text file of the manuscript (tex, txt, rtf, docx or doc), references, tables (including captions) and figure captions. Do not upload a PDF as your "Main Document".
- 2) A separate electronic file of each figure (EPS or print-quality PDF preferred (either format should be produced directly from original creation package), or original software format)
- 3) Included a 100 word media summary of your paper when requested at submission. Please ensure you have entered correct contact details (email, institution and telephone) in your user account

4) Included the raw data to support the claims made in your paper. You can either include your data as electronic supplementary material or upload to a repository and include the relevant doi within your manuscript

5) All supplementary materials accompanying an accepted article will be treated as in their final form. Note that the Royal Society will neither edit nor typeset supplementary material and it will be hosted as provided. Please ensure that the supplementary material includes the paper details where possible (authors, article title, journal name).

Kind regards,
Dr Ellis Wilde
Publishing Editor, Journals

On behalf of the Subject Editor Professor Anthony Stace and the Associate Editor Dr Dattatray Late.

RSC Associate Editor
Comments to the Author:
Accept with minor revisions

RSC Subject Editor
Comments to the Author:
(There are no comments.)

Reviewer comments to Author:

Reviewer: 1

Comments to the Author(s)

The manuscript has been revised mostly according to the comments and suggestions of the reviewers, I recommend the acceptance of the manuscript for publication in Royal Society Open Science.

Reviewer: 2

Comments to the Author(s)

The authors have addressed all my questions except the statistical analysis in figure 8b. After adding the statistical analysis, this work could be published in Royal Society Open Science.

Author's Response to Decision Letter for (RSOS-211004.R1)

See Appendix B.

Decision letter (RSOS-211004.R2)

Dear Dr Liu:

Title: Chemodrug-gated mesoporous nanoplatform for NIR light controlled drug release and synergistic chemophothermal therapy of tumors
Manuscript ID: RSOS-211004.R2

It is a pleasure to accept your manuscript in its current form for publication in Royal Society Open Science. The chemistry content of Royal Society Open Science is published in collaboration with the Royal Society of Chemistry.

Yours sincerely,
Ellis Wilde
Publishing Editor, Journals

On behalf of the Subject Editor Professor Anthony Stace and the Associate Editor Dr Dattatray Late.

RSC Associate Editor
Comments to the Author:
Accept as is

Reviewer(s)' Comments to Author:

s

Appendix A

Dear Editor,

We are submitting our revised manuscript entitled “**Chemodrug-gated mesoporous nanoplatform for NIR light controlled drug release and synergistic chemophothermal therapy of tumors**” to be considered for publication in **Royal Society Open Science**.

We thank you very much for serving as editor of our manuscript and also thanks very much for the referees’ comments and suggestions. The professional comments are all valuable and helpful for revising and improving our manuscript, as well as have important guiding significance to our researches. Now we have carefully addressed the comments and made the revision accordingly (red marked in the revision). We hope that you and the referee will now deem our manuscript suitable for publication in **Royal Society Open Science**. We have addressed the reviewer’s comments one by one behind this letter (blue marked).

Yours truly,

Correspondence to:

Dr. Jinliang Liu, PhD.

School of Environmental and Chemical Engineering

Shanghai University, 99 Shangda Road, 200444, Shanghai, China

Tel: +86-138-1633-5415

E-mail: liujl@shu.edu.cn

Response to Editors and Reviewers' Comments

Reviewers' comments:

Reviewer #1:

In this manuscript, the authors developed a multifunctional nanoplatform (GNR@mSiO₂-DOX/PFP@PDA) based on gold nanorods (GNR) coated with mesoporous silica (mSiO₂) for synergistic chemo-photothermal therapy of tumors. Doxorubicin hydrochloride (DOX) and perfluoro-n-pentane (PFP) were loaded into the pores of mSiO₂. A polydopamine (PDA) layer as the gatekeeper of pores was coated on their surface. The whole nano-system is too complex, and the novelty of this work is very poor. There are a lot of similar published works, this manuscript is not suitable for publication on Royal Society Open Science.

Authors' response: Thanks for your comments. In drug delivery research, the use of stimulation-responsive drug delivery to achieve more accurate and efficient release of drugs is widely favored. However, the release rate of pH, photothermal and other stimulating drug release methods still cannot achieve the desired effect. The nano-system we designed can use the bubbles generated by the phase change agent under the photothermal action to promote and accelerate the drug release together with the photothermal and pH action. We believe the work here does do contributions to the field of accelerating drug release and shortening treatment time.

1. What is the function of PFP in this nano-system?

Authors' response: Thanks for your questions. PFP is perfluoro-pentane, which can be gasified to generate bubbles when heated to high temperature. The original boiling point of PFP is 29 °C, however, it will rise to 40–50 °C due to the blood pressure after intravenous injection. [1] It makes use of the heat produced by the photothermal effect of gold nanorods in the nano-system to realize phase change and generate bubbles, which further accelerate the release of drugs from the pores of the nano-system.

2. In Figure 3a and b, one cannot see the difference within different concentrations of GNR solutions and GNR@mSiO₂ solutions. These experiments should redo.

Authors' response: Thanks for your kind suggestions and comments. We have redone these experiments. The change of temperature is mainly caused by the heat generated by gold nanorods (GNR) when stimulated by 808 nm excitation. The temperature change trends of the GNR solution and GNR@mSiO₂ solution with different concentrations under 808 nm laser irradiation at the power density of 2 W cm⁻² for 7 min are shown in **Fig.3a** and **Fig.3b** in the revised manuscript. (See Page 23) As the concentration of GNR increases, the temperature of the solution also changes with the increase. GNR@mSiO₂ represents the mesoporous SiO₂ modified GNR, on which the surface of GNR is coated with mesoporous SiO₂, so GNR@mSiO₂ solution and GNR solution have similar trends under this condition.

Fig.3a. Temperature elevation curves of GNR solutions with different concentrations under 808 nm laser irradiation at the power density of 2 W cm⁻² for 7 min.

Fig.3b. Temperature elevation curves of GNR@mSiO₂ solutions with different concentrations under 808 nm laser irradiation at the power density of 2 W cm⁻² for 7 min.

3. What is the purpose of microbubbles generation of PFP-GNR@mSiO₂?

Authors' response: Thanks for your questions. the purpose of microbubbles generation of PFP-GNR@mSiO₂ is to promote and accelerate the release of drug (DOX) loaded in the mesoporous SiO₂ channels under the photothermal effects of GNR.

4. The error bars are missing in Figure 4d.

Authors' response: Thanks for your comment. We conducted data processing and analysis on multiple sets of such data again, and the errors of drug quantities released at different times were expressed in **Figure 4d** in the revised manuscript. (See Page 24)

Fig. 4d. DOX release profiles from GNR@SiO₂-DOX and PFP@GNR@SiO₂-DOX with NIR laser irradiation (power density of 2 W cm⁻²).

5. The in vivo chemo-photothermal therapeutic effect is very poor, but the authors claimed that 'the tumor could be basically completely eliminated after 8 ~ 10 days (Figure 8b).' The results are inconsistent with Figure 8c.

Authors' response: Thank you for your question, and we have revised the discussion section of in vivo chemo-photothermal therapy. We have deleted the description "the tumor could be basically completely eliminated after 8 ~ 10 days", because the tumor volume is decreased but not disappeared. Compared with the control group (PBS), the tumor inhibitory ratio in GNR@mSiO₂-DOX/PFP@PDA + NIR group is estimated to be about 79%. In Figure 8c, the typical photographs of excised tumors also indicate that the treated mouse has the lowest tumor size.

Reviewer #2:

The present study provides a NIR light controlled multifunctional nanoplatform based on gold nanorods coated with mesoporous silica for synergistic chemo-photothermal therapy of tumors. The manuscript is well managed and the work is interesting. However, further work should be done and the paper should be minor revised

according to the following points before its acceptance.

Authors' response: We appreciate the reviewer's comments.

1. The authors claimed that the photothermal conversion of GNR can also lead the liquid-gas phase transition of PFP to generate the bubbles to accelerate the release of DOX. For this reason, the control group without NIR in Figure 4c should be added.

Authors' response: Thanks very much for your kind suggestions. The bubble formation of PFP@GNR@mSiO₂ nanomaterials in the presence or absence of near-infrared laser irradiation are shown in Figure 4c in the revised manuscript (See Page 24). In the absence of NIR, GNR does not generate heat, so there are basically no bubbles. Under the irradiation of near-infrared light, the photothermal effect of GNRs promoted the phase transformation of PFP, and the number of bubbles also increased with the increase of material concentration.

Fig. 4c. Number of microbubbles in PFP@GNR@mSiO₂ solutions with NIR laser irradiation (power density of 2 W/cm²) for 5 min or without NIR.

2. To evaluate the in vitro cellular cytotoxicity, it is suggested that the cell viability in normal cells should also be investigated, and the irradiation power of NIR in Figure 5b should also be mentioned.

Authors' response: Thanks for your comment. In the revised manuscript, we use BALB/3T3 normal cells to evaluate the in vitro cellular cytotoxicity, and the irradiation power, irradiation time of NIR are also mentioned in the revised

manuscript. (See Page 4, Figure S6 in the Supplementary Information)

Fig. S6. Cell cytotoxicity of normal cells without NIR or with NIR (power density of 2 W/cm^2) for 5 min.

3. For in vivo study of tumor growth curves in figure 8d, statistical analysis should be added.

Authors' response: Thank you for your question, and we have revised the discussion section of in vivo chemo-photothermal therapy. As shown in figure 8b, the tumor volume of the mice in groups (PBS, GNR@mSiO₂, GNR@mSiO₂-DOX@PDA, GNR@mSiO₂-DOX/PFP@PDA,) apparently increase over time while the tumor growth in groups (PBS + NIR, GNR@mSiO₂ + NIR, GNR@mSiO₂-DOX@PDA + NIR, GNR@mSiO₂-DOX/PFP@PDA + NIR) are greatly inhibited. Compared with the control group (PBS), the tumor inhibitory ratio in GNR@mSiO₂-DOX/PFP@PDA + NIR group is estimated to be about 79%. The typical photographs of excised tumors (Figure 8c) also indicate that the treatment of GNR@mSiO₂-DOX/PFP@PDA with 808 nm irradiation dramatically inhibits the tumor growth and the treated mouse has the lowest tumor size.

4. The manuscript contains some grammar errors and inconsistent formats in references section, please check them carefully.

Authors' response: Thanks for your corrections. We have corrected the grammatical errors and inconsistent formats in the original text.

5. There are some writing mistakes which should be carefully examined and revised. For example, in Figure 3d “ $R_2 = 0.998$ ” should be “ $R^2 = 0.998$ ”.

Authors' response: Thanks for your corrections. We have made changes to the error in the figure 3d in the revised manuscript. (See Page 23)

Fig.3d. Calculation of the time constant for heat transfer from the system using linear regression of the cooling profile.

References

[1] Li, C., Zhang, Y., Li, Z., Mei, E., Lin, J., Li, F., Chen, C., Qing, X., Hou, L., Xiong, L., et al. 2018 Light-Responsive Biodegradable Nanorattles for Cancer Theranostics. *Advanced Materials*. 30, (10.1002/adma.201706150)

Appendix B

Dear Editor,

We are submitting our revised manuscript entitled “**Chemodrug-gated mesoporous nanoplatform for NIR light controlled drug release and synergistic chemophothermal therapy of tumors**” to be considered for publication in **Royal Society Open Science**.

We thank you very much for serving as editor of our manuscript and also thanks very much for the referees’ comments and suggestions. The professional comments are all valuable and helpful for revising and improving our manuscript, as well as have important guiding significance to our researches. Now we have carefully addressed the comments and made the revision accordingly (red marked in the revision). We hope that you and the referee will now deem our manuscript suitable for publication in **Royal Society Open Science**. We have addressed the reviewer’s comments one by one behind this letter (blue marked).

Yours truly,

Correspondence to:

Dr. Jinliang Liu, PhD.

School of Environmental and Chemical Engineering

Shanghai University, 99 Shangda Road, 200444, Shanghai, China

Tel: +86-138-1633-5415

E-mail: liujl@shu.edu.cn

Response to Editors and Reviewers' Comments

Reviewers' comments:

Reviewer #1:

The manuscript has been revised mostly according to the comments and suggestions of the reviewers, I recommend the acceptance of the manuscript for publication in Royal Society Open Science.

Authors' response: We appreciate the reviewer's comments.

Reviewer #2:

The authors have addressed all my questions except the statistical analysis in figure 8b. After adding the statistical analysis, this work could be published in Royal Society Open Science.

Authors' response: Thanks for your kind suggestions and comments. We have revised the statistical analysis in figure 8b. The results of weight changes and tumor volume changes in each mouse were expressed as the mean \pm standard deviation (SD). Comparisons among the groups were evaluated by t-test (two variables) or ANOVA (multiple variables) followed by the Student-Newman-Keuls test. A value of $p < 0.05$ was considered statistically significant. As shown in figure 8b, the tumor volume of the mice in groups (PBS, PBS + NIR, GNR@mSiO₂, GNR@mSiO₂-DOX@PDA, GNR@mSiO₂-DOX/PFP@PDA,) apparently increase over time, and the tumor volume were about 4.5 times larger than the initial value, indicating that NIR irradiation alone and the nanosystem (GNR@mSiO₂, GNR@mSiO₂-DOX@PDA, GNR@mSiO₂-DOX/PFP@PDA) without NIR irradiation did not have effective prohibition ability on tumor growth. On the contrast, the tumor growth in groups (GNR@mSiO₂ + NIR, GNR@mSiO₂-DOX@PDA + NIR, GNR@mSiO₂-DOX/PFP@PDA + NIR) are greatly inhibited. Compared with the control group (PBS), the tumor inhibitory ratio in GNR@mSiO₂-DOX/PFP@PDA + NIR group is estimated to be about 79%. A value of $p < 0.001$ was obtained, which was considered statistically significant. (See page 14-15 and Figure 8b in the revised manuscript.)